

SciPost Phys. 1(1), 002 (2016)

# Tapered amplifier laser with frequency-shifted feedback

**A. Bayerle[1*], S. Tzanova[2,3], P. Vlaar[1], B. Pasquiou[1] and F. Schreck[1,†]**

**1** Van der Waals-Zeeman Institute, Institute of Physics, University of Amsterdam, 1098 XH Amsterdam, Netherlands.
**2** Institut für Quantenoptik und Quanteninformation, Österreichische Akademie der Wissenschaften, 6020 Innsbruck, Austria.
**3** Institut für Experimentalphysik, Universität Innsbruck, 6020 Innsbruck, Austria.

* a.bayerle@uva.nl
† http://www.strontiumbec.com/

## Abstract

We present a frequency-shifted feedback (FSF) laser based on a tapered amplifier. The laser operates as a coherent broadband source with up to 370 GHz spectral width and 2.3 $\mu$s coherence time. If the FSF laser is seeded by a continuous-wave laser a frequency comb spanning the output spectrum appears in addition to the broadband emission. The laser has an output power of 280 mW and a center wavelength of 780 nm. The ease and flexibility of use of tapered amplifiers makes our FSF laser attractive for a wide range of applications, especially in metrology.


# 1 Introduction

An FSF laser consists of a pumped gain medium inside a cavity that also contains a frequency shifting element such as an acousto-optic modulator (AOM). After each round trip the photons experience an increment in frequency through the AOM. Consequently exponential amplification of single frequency modes will not occur, since the fields of different round trips through the cavity cannot interfere constructively. Instead, spontaneously emitted photons from the gain medium are frequency shifted and amplified, resulting in a coherent modeless broadband emission spectrum. Seeding the FSF laser with a continuous-wave (cw) laser adds a second spectral component to the laser output, a comb of equidistantly spaced frequency components [1]. If parameters are chosen correctly, the FSF laser can generate Fourier-limited laser pulses with tunable repetition rate [2–4].

FSF lasers have many applications. Their coherent broadband emission is used for optical domain ranging, as in [5], where a distance of 18.5 km was measured with parts-per-million accuracy. The broad spectrum of FSF lasers allows one to cover a larger velocity class in Doppler broadened atomic transitions, improving optical pumping of room temperature He* gases [6] or increasing the fluorescence in Na vapors, which could improve astronomy guide star techniques [7]. The frequency comb like feature of an externally seeded FSF laser can be applied to spectroscopy, wavelength division multiplexed coherent communications [8] and optical frequency referencing [9].

Several implementations of FSF lasers with different gain media were reported. Early versions were based on HeNe [10], dye [11, 12], Ti:sapphire [3], and Nd:YLF lasers [4]. Modern implementations make use of a $Yb^{3+}$ doped fiber amplifiers [13] as well as semiconductor lasers [14–17].

In this article we demonstrate an FSF laser that uses a tapered amplifier (TA) as gain medium. TAs are electrically pumped semiconductor laser amplifiers, which are capable of emitting single-mode laser radiation at Watt levels. TA chips are commonplace in many laboratories and can be built into miniaturized setups [18]. The relatively cheap devices are available at a wide range of emission wavelengths. Our work leverages these advantages, making FSF lasers easily accessible to more researchers, thereby widening the range of applications of these light sources. In mutually independent work, a similar system was developed for broadband laser cooling, as indicated by conference abstract [19]. In the following, we present first our experimental setup and then the spectral properties of the FSF laser.

# 2 Experimental setup

The experimental setup consists of four electro-optical sub-systems shown in Fig. 1a-d, namely the FSF laser (a), the seed source (b), and the characterization tools (c,d). Each sub-system will now be described in detail. Our FSF laser consists of a TA and an AOM inside a four-mirror ring-cavity of 1440 mm optical length. The total loss through the mirrors is 1.4%. The gain medium of the laser is realized by a tapered amplifier (Toptica TA-0780-2000-4). Its gain profile is centered at 780 nm with a full width at half maximum (FWHM) of 10 nm. The TA chip is temperature stabilized to 20 °C and operated at a current of 1 A yielding an output power of 300 mW. The divergent output beam of the TA is collimated by the lens $L_{TA2}$. The Faraday isolator (FI) prevents laser light from traveling in the backwards direction, which would damage the input facet of the TA. The emission of the TA is split into two orthogonally polarized parts by a half-wave plate and a polarizing beam splitter (PBS). The vertically polarized reflection off the PBS serves as the output port of the FSF laser which is sent to the diagnostic tools via optical fiber. The horizontally polarized transmission of the PBS continues into the feedback

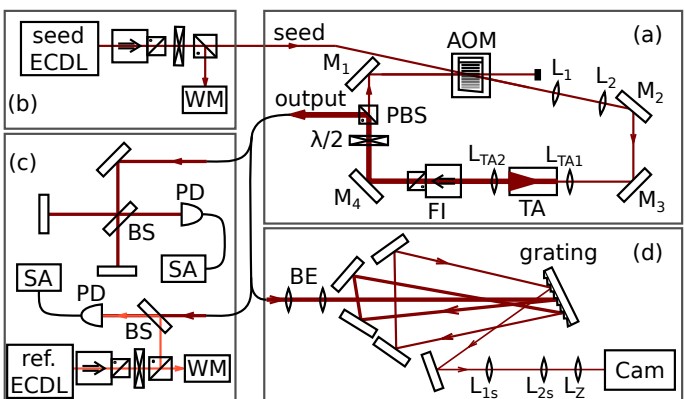

Figure 1: Schematic of the FSF laser and characterization tools. **(a)** cavity of the FSF laser, **(b)** seed laser, **(c)** radio frequency characterization setup, **(d)** grating spectrometer.

loop. The power in the feedback loop is deduced from the power of the leakage light through mirror $M_1$. The half-wave plate is adjusted such that the power at the TA input facet remains below 30 mW in order to avoid damaging the TA chip. The feedback light propagating through the acousto-optic modulator (AOM, Gooch & Housego 3080-120 or Crystal Technology 3350-190) is shifted in frequency by $f_{AOM} = 60$ to 380 MHz with efficiencies of up to 87%. Lenses $L_1$ and $L_2$ adapt the beam waist for optimal injection of the TA through focusing lens $L_{TA1}$. The focal distance of $L_{TA1}$ and the orientation of mirrors $M_2$, $M_3$ is adjusted to achieve maximal output power of the TA.

The seed laser light (Fig. 1b) is provided by an external cavity diode laser (ECDL, Toptica DL Pro). The ECDL single-mode emission is tunable around 384.25 THz (780.2 nm) with a linewidth below 1 MHz on a timescale of 5 $\mu$s. The ECDL optical frequency is measured by a Burleigh 1500 wavemeter (WM) with 0.5 GHz precision. The seed laser beam is introduced into the FSF ring cavity through the back of the AOM coaligned with the first order diffraction of the feedback.

The spectral properties of the output of the FSF laser are analysed in the optical and the RF domain. For phase noise measurements in the radio frequency (RF) domain the output of the FSF laser is sent through a Michelson interferometer shown in the upper half of Fig. 1c. The light from the output port of the interferometer is detected by a 2 GHz bandwidth photodiode (PD) connected to a spectrum analyzer (SA, Rohde-Schwarz FSH8). A second interferometer setup, used to resolve the mode structure of the FSF laser, is shown in the lower part of Fig. 1c. The output is coaligned with the equally polarized beam of a second ECDL laser on a PD, producing a beat signal between the two lasers. The beat signal is measured by the SA and the optical frequency of the reference ECDL is measured by the Burleigh 1500 wavemeter (WM).

The optical spectrum of the FSF laser output is recorded by a spectrometer (Fig. 1d). The dispersive element of the spectrometer is a holographic grating with 1800 grooves per mm and a size of 50 mm×50 mm. In order to illuminate a large number of grooves and increase the resolution of the spectrometer, the input beam is widened by a beam expander (BE) to roughly 20 mm. The first order diffraction is guided back onto the same grating at a different angle by a pair of mirrors. The same arrangement is repeated so that the resolution is enhanced three-fold. The third diffraction off the grating is sent through a telescope ($L_{1s}$, $L_{2s}$) to decrease the beam waist to 1 mm. Subsequently the diffracted beam is projected onto a CMOS camera (Cam) by a cylindrical lens of 150 mm focal length. The spectrometer is calibrated using the seed laser set to different optical frequencies. The measured resolution of the spectrometer is 4 GHz.

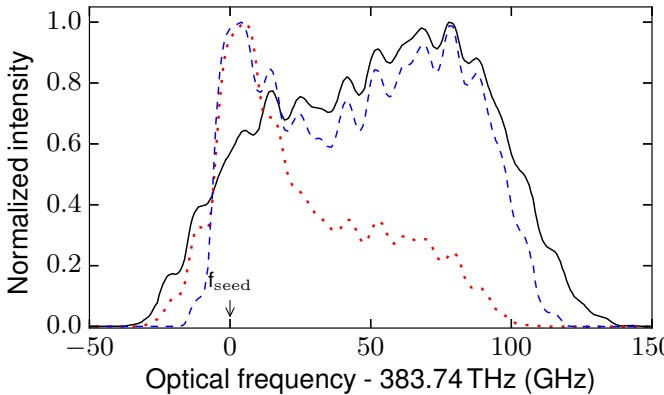

Figure 2: Optical spectrum of the FSF laser measured with the grating spectrometer without external seed (solid black) and an external seed power of $460\,\mu$W (dashed blue) and $720\,\mu$W (dotted red).

## 3  Properties of the laser

Two modes of operation of the frequency-shifted feedback laser are realized. Firstly, the laser is internally seeded by spontaneous emission from the tapered amplifier, resulting in the emission of modeless broadband light. Secondly, the cw-laser seed light is introduced into the cavity resulting in a comb of narrow, equidistant frequency components in addition to the modeless emission.

### 3.1  Broadband modeless laser

We first discuss the operation of the laser without external seed. We use $f_{AOM} = 80\,$MHz and chose the $+1$-order diffracted beam of the AOM for feedback. The diffraction efficiency is 75%. After the feedback is injected into the TA the output power of the FSF laser reaches 280 mW, which matches the specified output power of the TA.

The optical spectrum of the FSF laser is shown in Fig. 2. The center of the profile is tunable over 1 THz (2 nm) by changing the angle of the injection mirror $M_2$ in the horizontal plane and here is set to 383.0 THz (782.75 nm). The full width half maximum of the spectrum is 120 GHz, independent of the center frequency. The spectrum can be broadened by increasing the AOM frequency, as shown in Sec. 3.2.

To prove that the broadband laser emission is modeless we use the beat setup (Fig. 1c). The reference ECDL frequency is set close to the center of the FSF output spectrum and scanned over 200 GHz. The measurement reveals an RF spectrum without beat signals as expected for a modeless spectrum.

The RF spectrum obtained through the Michelson interferometer (see Fig. 3a) also shows the characteristic features of a modeless broadband laser. The spectrum exhibits a comb structure with lines separated by integer multiples of the cavity free spectral range ($n \times f_{FSR}$), where each comb line is accompanied symmetrically by a pair of lines at $n \times f_{FSR} \pm f_b$. The comb structure stems from the fact that the initial spontaneous emission of the TA reoccurs after each round trip [1] and has a spacing of $f_{FSR} = 208.54(6)\,$MHz. The frequency difference between the comb lines and the side peaks, $f_b$, is related to the arm length difference $\Delta$L of the Michelson interferometer and $f_b = \gamma 2 \Delta L/c$, where $\gamma = f_{AOM} f_{FSR} = 1.6683(3) \times 10^{16}\,$Hz/s is called the chirp rate of the FSF laser [20]. The measured value $f_b = 28.95(6)\,$MHz corresponds to an arm length difference of 260.12(5) mm. This value is consistent with a direct length measurement using a ruler, demonstrating the use of our FSF laser as range finder [20]. It was shown

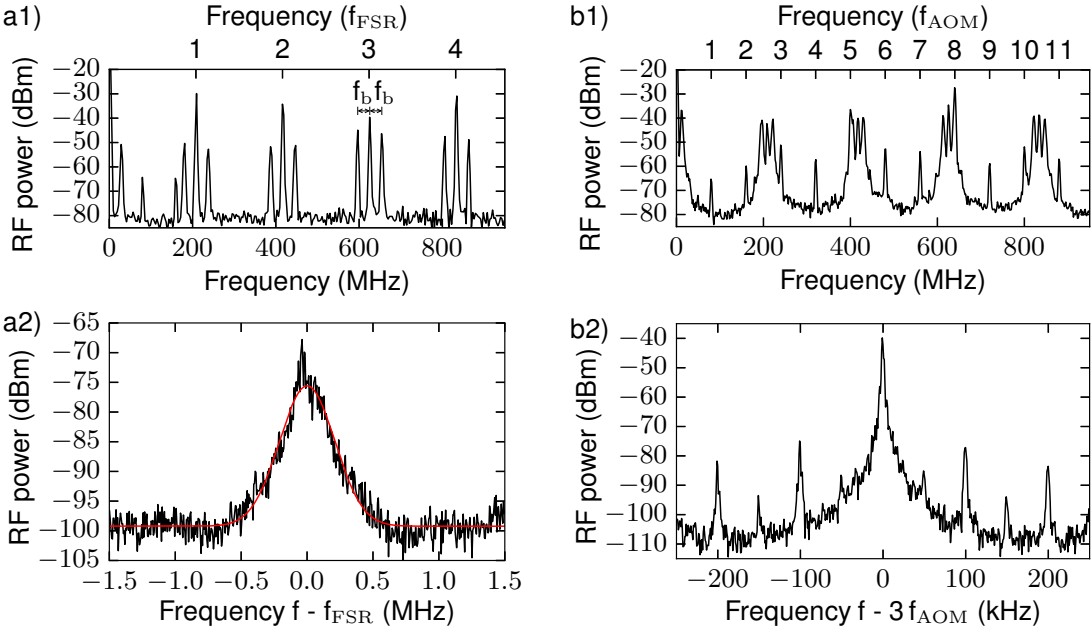

Figure 3: RF noise spectrum of the Michelson interferometer output. **(a)** Output of the modeless broadband laser. **(a2)** Zoom on the peak at $f_{\mathrm{FSR}}$ and Gaussian fit. The width of the Gaussian is 430(7) kHz. **(b)** Output of the FSF laser seeded with 450 $\mu$W cw narrow-band light. **(b2)** Zoom on the peak at $3f_{\mathrm{AOM}}$. The half-width at half-maximum is (30±2) Hz.

in [1] that the peaks in the RF noise spectrum are of Gaussian shape with a width related to the effective photon lifetime inside the cavity. Figure 3a2 shows the peak at $f_{\mathrm{FSR}}$ with higher resolution. The FWHM of the peak is 430(7) kHz and hence the lifetime of a photon in the cavity is $t_{\mathrm{coh}} = 2.33(4)\,\mu s$.

## 3.2 Externally seeded laser

In the experiments described so far spontaneous emission is the only source of seed for the FSF laser. Now the laser is seeded with a small amount of narrow-band cw radiation. The cw seed introduces a mode with fixed frequency into the ring cavity from which the amplification and frequency shifting process starts. A sequence of comb lines alongside the modeless broadband emission of the FSF laser is expected to appear [1], which we demonstrate experimentally in the following. Since the AOM shifts light to higher frequencies, the seed laser frequency is set to a value close to the low-frequency edge of the output spectrum of the FSF laser without external seed. The spectrum changes compared to the case without external seed, as shown in Fig. 2 for two seed powers. For 460 $\mu$W seed power a sharp increase in intensity appears at the frequency of the seed laser and extending 50 GHz to higher frequencies. For even higher frequencies the spectrum is similar to the spectrum of the modeless FSF laser. If the laser is seeded with higher power (720 $\mu$W, red curve), the feature near the seed frequency decays faster, and the lobe at higher frequencies shrinks substantially and the overall width decreases.

Compared to the situation without external seed, the RF spectrum at the output port of the Michelson interferometer (Fig. 3b) contains additional sharp lines at integer multiples of $f_{\mathrm{AOM}}$. Figure 3b2 shows a high-bandwidth measurement of such a line at $3f_{\mathrm{AOM}}$. The FWHM is less than 100 Hz and therefore narrower than the broadband emission peak shown in Fig. 3a2

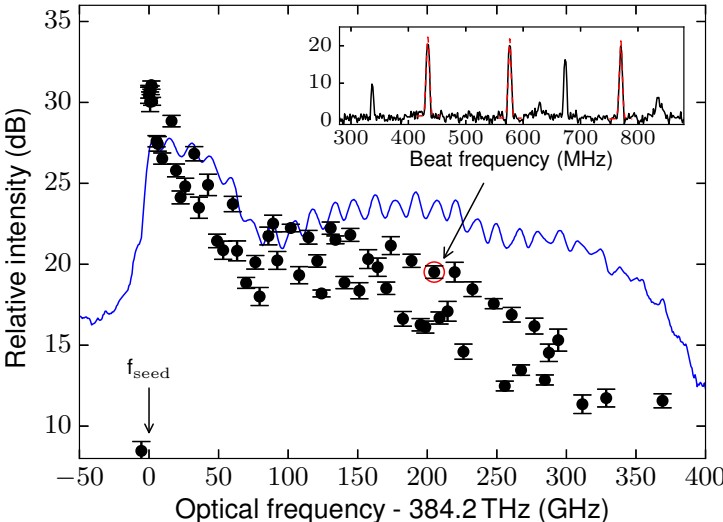

Figure 4: Spectrum of the seeded FSF laser. The intensity within a few GHz bandwidth is given relative to an arbitrary reference value. The blue trace stems from the grating spectrometer. Black circles show the height of the beat signal between the reference laser and FSF output, relative to the background. The inset shows the RF spectrum corresponding to the data point marked by an arrow and being encircled.

by four orders of magnitude.

Figure 4 shows the spectrum of a frequency comb spanning 370 GHz. To obtain a frequency comb over this larger spectral width compared to before we find it necessary to increase the AOM frequency and diffraction efficiency. In the following we work with $f_{AOM} = 370$ MHz and optimize the diffraction efficiency to 87%. The power of the seed laser at the TA input facet is set to 80 $\mu$W. The mode structure of the cw seeded FSF laser is resolved by means of the beat measurement with the reference ECDL. The inset shows an example RF spectrum of the beat photodiode signal. Every comb line at $f_n = f_{seed} + n \times f_{AOM}$ of the FSF laser beats with the reference laser at frequency $f_{ref}$, leading to peaks (dashed red) at $|f_n - f_{ref}|$. The other features in the inset are beat signals between comb lines at $m \times f_{AOM}$ and more noisy peaks at multiples of $f_{FSR}$. The reference laser frequency $f_{ref}$ is now incremented in 5 GHz steps. The height of the strongest beat signal between the reference laser and the FSF output in a range from DC to 2 GHz is plotted against the optical frequency of the reference laser in Fig. 4. We observe that the spectral weight of the frequency comb decreases roughly exponentially with frequency. For comparison we also show the spectrum obtained by the grating spectrometer in Fig. 4 (solid blue). Since this spectrum shows the sum of the modeless broadband spectrum and the frequency comb, we observe that the ratio of broadband intensity to comb intensity increases with higher frequency. This behavior is a consequence of the accumulation of amplified spontaneous emission for higher frequencies.

Although the presence of a narrow-band cw seed adds a sequence of equidistant modes to the spectrum, the signature of broadband emission is present in the spectrum for all seed intensities explored here. In contrast to a conventional laser where a single mode can dominate the spectrum due to exponential amplification [21], in an FSF laser single modes cannot be favoured. The frequency shift in each round trip does not allow for constructive interference. Instead spontaneous emission as well as a cw seed is amplified in the FSF laser. Therefore the discussed spectral features, the frequency comb and the broadband emission, coexist in the laser when seeded with a cw source [1]. To characterize the relative importance of the two spectral features, we measure the intensity of the corresponding RF peaks (see Fig. 3a2 and

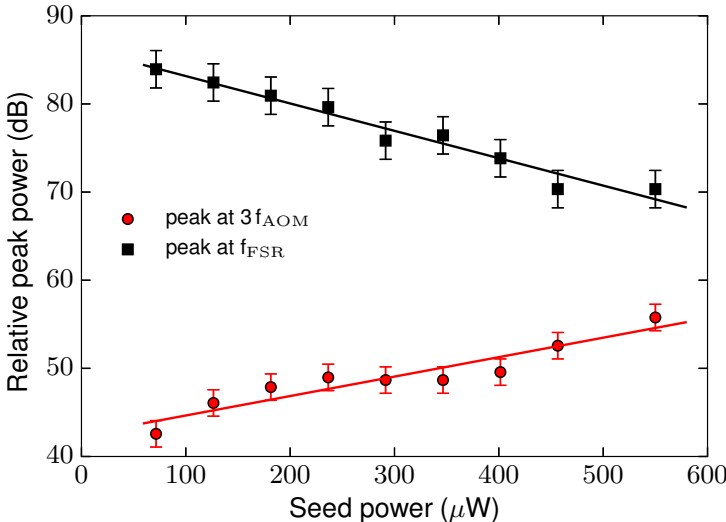

Figure 5: Dependence of the broadband signal at the frequency $f_{FSF}$ (black squares) and comb feature at $3f_{AOM}$ (red circles) on seeding laser power and linear fits to the data. The relative peak power is obtained by integrating peaks such as the ones shown in Fig. 3a2 and 3b2 over frequency and comparing them to the noise floor.

3b2), in dependence of the seed power. For each seed power the power of the light circulating in the cavity is adjusted to a safe, fixed value using the intracavity half-wave plate. We record the height of the RF peaks relative to the background at $n \times f_{FSR}$ and $m \times f_{AOM}$. The averaged values of each type of frequency component for m, n = 1 to 5 are shown in Fig. 5. Within the range of parameters examined the comb intensity increases exponentially with seed power, whereas the broadband emission is exponentially suppressed.

The comb can be stabilized in frequency by locking the seed laser to a desired frequency, e.g. a Rb spectroscopy line. In previous work such stabilization required an additional lock of a comb line to a reference laser, see [13, 22] and references therein.

## 4 Conclusion and outlook

We have demonstrated an FSF ring laser based on a tapered amplifier. The FSF laser emits coherent modeless broadband radiation with 120 GHz bandwidth and a coherence time of 2.3 µs. The center frequency of the broadband source is tunable over 1 THz. A frequency comb spanning up to 370 GHz was realized by seeding the FSF laser with a narrow-band cw source. Both spectral components existed simultaneously in the FSF laser and we measured their relative strength depending on seed power.

Since TAs are relatively cheap devices, available for many wavelengths, we expect that our work makes FSF lasers easily accessible to more researchers, leading to more applications of these devices. In contrast to other frequency combs [8, 23] our comb has a narrower spectrum. The comb covers 0.7 nm, which is a fraction of the 3dB gain bandwidth of our TA (11 nm). We observe that the spectrum can be broadened by increasing the AOM frequency, although less than proportional to that frequency. Broadening the comb width would be valuable to increase the range of applications of our scheme and could be a topic for further research. Still, a comb spanning hundreds of GHz is relevant to many applications and we plan to use the frequency comb of our FSF laser as frequency reference for photoassociation spectroscopy of RbSr molecules [24].

## Acknowledgements

We thank R. Grimm for generous support. We gratefully acknowledge funding by the European Research Council (ERC) under Project No. 615117 and by the Austrian Ministry of Science and Research (BMWF) and the Austrian Science Fund (FWF) through a START grant under Project No. Y507-N20. B.P. thanks the NWO for funding through Veni grant No. 680-47-438.

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
