# Peer review of "Tapered amplifier laser with frequency-shifted feedback"

_SciPost Physics, doi:SciPost Phys. 1, 002 (2016)_

## Round 2 · Referee Report · Anonymous · 2016-9-8

Strengths
It is clearly written and instructive.
The first tapered amplifier with frequency shift feedback.
The technique is a relatively inexpensive and straightforward technique to generate a frequency comb.
Weaknesses
The one main drawback of the technique is the relatively small frequency range available in comparison to other frequency combs.
Report
The manuscript “Tapered amplifier laser with frequency-shifted feedback” is clearly written and instructive. It details the first tapered amplifier with frequency shift feedback (although I think the authors may wish to reference the abstract, http://adsabs.harvard.edu/abs/2014APS..MAR.C1323L – found via Google, in fairness).
The technique is a relatively inexpensive and straightforward technique to generate a frequency comb, which could then be e.g. frequency referenced to a rubidium saturated absorption spectrum, shown in e.g. dois: 10.1364/OE.22.010494, 10.1063/1.4801996, which may be worth referencing as well as mentioning in the paper.
The one main drawback of the technique is the relatively small frequency range available in comparison to other frequency combs, which the authors may wish to comment on. It's a shame that you can't reach the 10nm ~ 5THz bandwidth of the TA, and assume if there was a way you would have done it.
I would at some level argue that the 3dB range is the bandwidth of the comb - but appreciate giving a 20dB width is acceptable in the comb community (and the 10dB width is half of the 20dB width).
In conclusion, a clear and useful paper which should be published with minor corrections.
Requested changes
Reference the abstract, http://adsabs.harvard.edu/abs/2014APS..MAR.C1323L – found via Google, in fairness).
Frequency referencing to a rubidium saturated absorption spectrum is worth mentioning in the paper and referencing, e.g. dois: 10.1364/OE.22.010494, 10.1063/1.4801996.
Comment on the limitations in comparison to other frequency combs.
Minor changes/corrections:
P3 1MHz linewidth - on what timescale? rms?
P3 344.25 => 384.25THz
P3 Find and replace groves => grooves
P5 Figure 3 has seeded output at ref freq. (c,d) but not described in text til 3.2.
Figure 3 caption needs capitalisation for each section.
P5 “by orders of magnitude” => “by four orders of magnitude”
P7 as blue curve => as shown in the blue curve.
Fig 5 => Please check this is the total integrated power, and not the spectral power density in dB/Hz (if so please correct accordingly).
Alex Bayerle on 2016-10-14 [id 68]
We thank the referees for their helpful comments.\
Here is our response.\
\textbf{referee 1}\
Reference the abstract, http://adsabs.harvard.edu/abs/2014APS..MAR.C1323L found via Google, in fairness). In the last paragraph of Sec.~1, we changed
we demonstrate for the first time an FSF laser that'' to
we demonstrate an FSF laser that'' and added ``In mutually independent work, a similar system was developed for broadband laser cooling, as indicated by conference abstract [19].''.\Frequency referencing to a rubidium saturated absorption spectrum is worth mentioning in the paper and referencing, e.g. dois:10.1364/OE.22.010494, 10.1063/1.4801996. At the end of Sec.~2.3 we added ``The comb can be stabilized in frequency by locking the seed laser to a desired frequency, e.g. a Rb spectroscopy line. In previous work such stabilization required an additional lock of a comb line to a reference laser, see [13, 22] and references therein.''\
Comment on the limitations in comparison to other frequency combs. In Sec. 4 we added ``In contrast to other frequency combs [23, 8] our comb has a narrower spectrum. It covers 0.7\,nm, which is a fraction of the 3dB gain bandwidth of our TA (11\,nm). We observe that the spectrum can be broadened by increasing the AOM frequency, although less than proportional to that frequency. Broadening the comb width would be valuable to increase the range of application of our scheme and could be a topic for further research. Still, a comb spanning hundreds of GHz is relevant to many applications and we plan to use the frequency comb of our FSF laser as frequency reference for photoassociation spectroscopy of RbSr molecules [24].''\
P3 1\,MHz linewidth - on what timescale? rms?
a linewidth below 1\,MHz.''$\rightarrow$
a linewidth below 1MHz on a timescale of 5us.''\P3 344.25\,THz $\rightarrow$ 384.25\,THz done\
P3 Find and replace groves $\rightarrow$ grooves done\
P5 Figure 3 has seeded output at ref freq. (c,d) but not described in text until 3.2. We think that a reader should be able to see from the figure caption that Fig. 3b belongs to Sec. 3.2. If we would slit the figure the reader would have a harder time to compare Fig.3a and Fig.3b, which would be a loss. Shifting it to after the start of Sec.3.2 would or shifting it to after the start of Sec. 3.2 would force the reader to flip through more pages when discussing Fig.3a and shift all later figures away from the page on which they are referred to. We chose to leave the figure as is.\
Figure 3 caption needs capitalisation for each section. done\
P5
by orders of magnitude'' $\rightarrow$
by four orders of magnitude'' done\P7 as blue curve $\rightarrow$ as shown in the blue curve.
For comparison we also show the spectrum obtained by the grating spectrometer as blue curve in Fig.~4.''->
For comparison we also show the spectrum obtained by the grating spectrometer in Fig.~4 (solid blue).''\Fig 5 => Please check this is the total integrated power, and not the spectral power density in dB/Hz (if so please correct accordingly). Formerly indeed the spectral power density was given. We now integrated the signals over frequency. The vertical label is changed to
Relative peak power'' and in the caption we added
The relative peak power is obtained by integrating peaks such as the ones shown in Fig. 3a2 and 3b2 over frequency and comparing them to the noise floor.''.\\textbf{referee 2:}\
We do not have a full understanding of the properties and limitations of our spectrum. We tried to increase the spectrum and found experimentally that increasing the AOM efficiency and frequency helped. But the width of the comb is not proportional to the AOM frequency, it grows less strongly. We speculate that the increased amount of amplified spontaneous emission in a frequency band corresponding to the AOM frequency reduces the number of amplified roundtrips through the cavity. We clarify the relation of spectral width and AOM frequency by the following modifications.\
The full width half maximum of the spectrum is 120 GHz.'' $\rightarrow$
The full width half maximum of the spectrum is 120\,GHz independent of the center frequency. The spectrum can be broadened by increasing the AOM frequency, as shown in Sec.\,3.2.'' This points the reader to the already existing text ``Figure 4 shows the spectrum of a frequency comb spanning 370\,GHz. To obtain a frequency comb over this larger spectral width compared to before we find it necessary to increase the AOM frequency and diffraction efficiency. In the following we worked with f$_{\rm AOM}$ = 370\,MHz and optimized the diffraction efficiency to 87\%.''\Finally we tell the reader that more work is needed to understand the spectrum, especially its width, at the end of Sec.~4, see our response to ``Comment on the limitations in comparison to other frequency combs.'' above.\
\textbf{other changes:}\
Figure labels corrected (To do: list changes)\
The FSF laser exhibits coherent modeless broadband radiation''$\rightarrow$
The FSF laser emits coherent modeless broadband radiation''\``A. Bayerle$^{1,}$, S. Tzanova$^{2,3}$, P. Vlaar$^1$ B. Pasquiou$^1$ F. Schreck$^{1,y}$"$\rightarrow$"A. Bayerle$^{1,}$, S. Tzanova$^{2,3}$, P. Vlaar$^1$, B. Pasquiou$^1$, and F. Schreck$^{1,y}$"\
use of a Yb$^{3+}$ doped fiber amplifiers''$\rightarrow$
use of Yb$^{3+}$ doped fiber amplifiers"\P5: "an arm length difference of 26.0(4)\,cm."$\rightarrow$"an arm length difference of 26.00(4)\,cm.''\
added sentence on P6: ``The power of the seed laser at the TA input facet was 80$\mu$W.''\
P5:
Each sub-system will be now described in detail.''$\rightarrow$
Each sub-system will now be described in detail.''\P3:
used to resolve the mode structure of the FSF laser, is shown in Fig.~1c.''$\rightarrow$
used to resolve the mode structure of the FSF laser, is shown in the lower part of Fig.~1c.''\P6:
contains additional sharp lines at integer multiples of f$_{AOM}$''. $\rightarrow$
contains additional sharp lines at integer multiples of f$_{/rm AOM}$.''\P7:
is present in the spectrum for all seed intensities presented here.''$\rightarrow$
is present in the spectrum for all seed intensities explored here.''\P5:
Fig. 3a and 3b''$\rightarrow$
Fig. 3a2 and 3b2''\ P8:Fig. 6'' $\rightarrow$
Fig. 5''

---

## Round 2 · Referee Report · Anonymous · 2016-9-16

Strengths
This is a very nice paper. It is clearly written and provides all necessary details for other researchers to reproduce the results.
Weaknesses
No important weaknesses.
Report
The authors report on a frequency-shifted feedback laser system that they have realized using a tapered amplifier. Such laser systems have been implemented and studied for a long time, but usually they require rather complex experimental setups. In using a tapered amplifier as the gain medium for the first time, the authors make these laser systems much more accessible.
Requested changes
I only have one minor question: Is a qualitative understanding of the “global” properties of the output spectra in Figs. 2 and 3 possible? Am I e.g. correct in assuming that the width of the optical spectrum is mainly determined by (or at least roughly proportional to) the AOM frequency, plus some other smaller contributions? The authors list 370 GHz as the upper limit for this width in the seeded case, but over what range can this be changed (both in the seeded and unseeded case)?

---

## Round 3 · List of Changes

In the last paragraph of Sec. 1, we changed "we demonstrate for the first time an FSF laser that" to "we demonstrate an FSF laser that" and added "In mutually independent work, a similar system was developed for broadband laser cooling, as indicated by conference abstract [19].".

At the end of Sec. 2.3 we added "The comb can be stabilized in frequency by locking the seed laser to a desired frequency, e.g. a Rb spectroscopy line. In previous work such stabilization required an additional lock of a comb line to a reference laser, see [13, 22] and references therein."

In Sec. 4 we added "In contrast to other frequency combs [23, 8] our comb has a narrower spectrum. It covers 0.7 nm, which is a fraction of the 3dB gain bandwidth of our TA (11 nm). We observe that the spectrum can be broadened by increasing the AOM frequency, although less than proportional to that frequency. Broadening the comb width would be valuable to increase the range of application of our scheme and could be a topic for further research. Still, a comb spanning hundreds of GHz is relevant to many applications and we plan to use the frequency comb of our FSF laser as frequency reference for photoassociation spectroscopy of RbSr molecules [24]."

"a linewidth below 1MHz."->"a linewidth below 1MHz on a timescale of 5us."

- P3 344.25 => 384.25THz

- P3 replaced groves with grooves

- P5 "by orders of magnitude" => "by four orders of magnitude"
done

- P7 "For comparison we also show the spectrum obtained by the grating spectrometer as blue curve in Fig. 4."->"For comparison we also show the spectrum obtained by the grating spectrometer in Fig. 4 (solid blue)."

- Fig 5: We now integrated the signals over frequency. The vertical label is changed to "Relative peak power " and in the caption we added "The relative peak power is obtained by integrating peaks such as the ones shown in Fig. 3a2 and 3b2 over frequency and comparing them to the noise floor.".

P4: "The full width half maximum of the spectrum is 120 GHz."->"The full width half maximum of the spectrum is 120 GHz independent of the center frequency. The spectrum can be broadened by increasing the AOM frequency, as shown in Sec. 3.2."

P6: "The power of the seed laser at the TA input
facet was 80uW."

"The FSF laser exhibits
coherent modeless broadband radiation"->"The FSF laser emits
coherent modeless broadband radiation"

"A. Bayerle1*, S. Tzanova2,3, P. Vlaar1 B. Pasquiou1 F. Schreck1,y"->"A. Bayerle1*, S. Tzanova2,3, P. Vlaar1, B. Pasquiou1, and F. Schreck1,y"

P5: "an arm length difference of 26.0(4) cm."->"an arm length difference of 26.00(4) cm."

P5: "Each sub-system
will be now described in detail."->"Each sub-system
will now be described in detail."

P3: "used to resolve the mode structure of the FSF laser, is
shown in Fig. 1c."->"used to resolve the mode structure of the FSF laser, is shown in the lower part of Fig. 1c."

P6: "contains additional sharp lines at integer multiples
of f_{AOM}."->"contains additional sharp lines at integer multiples
of fAOM. contains additional sharp lines at integer multiples
of f_{/rm AOM}."

P7: "is present in the spectrum for all
seed intensities presented here."->"is present in the spectrum for all
seed intensities explored here."

P5: "Fig. 3a and 3b"->"Fig. 3a2 and 3b2"

P8: "Fig. 6" -> "Fig. 5"

---

## Editorial Decision

published